# Probing the decision-making mechanisms underlying choice between drug and nondrug rewards in rats

Youna Vandaele[1]*, Magalie Lenoir[2,3], Caroline Vouillac-Mendoza[2,3], Karine Guillem[2,3], Serge H Ahmed[2,3]*

[1]Lausanne University Hospital, Department of Psychiatry, Prilly, Switzerland; [2]Université de Bordeaux, Institut des Maladies Neurodégénératives, Bordeaux, France; [3]CNRS, Institut des Maladies Neurodégénératives, Bordeaux, France

**Abstract** Delineating the decision-making mechanisms underlying choice between drug and nondrug rewards remains a challenge. This study adopts an original approach to probe these mechanisms by comparing response latencies during sampling versus choice trials. While lengthening of latencies during choice is predicted in a deliberative choice model (DCM), the race-like response competition mechanism postulated by the Sequential choice model (SCM) predicts a shortening of latencies during choice compared to sampling. Here, we tested these predictions by conducting a retrospective analysis of cocaine-versus-saccharin choice experiments conducted in our laboratory. We found that rats engage deliberative decision-making mechanisms after limited training, but adopt a SCM-like response selection mechanism after more extended training, while their behavior is presumably habitual. Thus, the DCM and SCM may not be general models of choice, as initially formulated, but could be dynamically engaged to control choice behavior across early and extended training.

*For correspondence:
youna.vandaele@chuv.ch (YV);
serge.ahmed@u-bordeaux.fr
(SHA)

Competing interests: The authors declare that no competing interests exist.

## Introduction

Investigating the decision-making mechanisms underlying choice between drug and nondrug rewards is essential to understand how their alterations can contribute to substance use disorders. Over the past decade, a growing research effort has been made to model this choice situation in animals, particularly in rodents (*Lüscher et al., 2020*; *Ahmed, 2018a*; *Ahmed, 2018b*). Overall, when given a choice, most rats prefer the nondrug alternative (e.g., sweet water, food pellets or social interaction) over potent drugs of abuse, such as cocaine or heroin (*Cantin et al., 2010*; *Lenoir et al., 2013a*; *Lenoir et al., 2007*; *Caprioli et al., 2017*; *Caprioli et al., 2015*; *Venniro et al., 2018*; *Vandaele et al., 2016*). Only few individual rats prefer the drug. This pattern of individual preferences is observed even after extended drug use and regardless of the drug dose available (*Cantin et al., 2010*; *Lenoir et al., 2007*). We recently found that, after some amount of training in our procedure, preference may not necessarily involve a comparison and deliberation over options' value, but could instead rely on more automatic processes such as habits (*Vandaele et al., 2020*; *Vandaele et al., 2019a*). However, investigating habitual control in a drug choice setting is particularly challenging, mainly because there is no effective method of drug reward devaluation in animals, particularly for drugs administered intravenously. An alternative approach, sufficiently versatile to probe the decision-making mechanisms underlying individual preferences in a drug choice setting, is thus needed.

Previous research aimed at testing the validity of different models of animal choice, including behavioral ecology-inspired models, have proposed that the decision-making mechanisms underlying choice should be, in theory, inferable from a detailed analysis of response latencies

(*Kacelnik et al., 2011*; *Monteiro et al., 2020*; *Shapiro et al., 2008*; *Vasconcelos et al., 2010*). One such model, the deliberative choice model (DCM), makes the assumption that deliberation requires time. Indeed, although deliberative behavior is highly flexible in adapting to new environmental contingencies, weighting the pros and cons of multiple choice outcomes to select the most valuable option requires higher cognitive demand and is therefore time-consuming (*Keramati et al., 2011*; *Wickelgren, 1977*; *Gold and Shadlen, 2007*). Specifically, if choice behavior involves a deliberation over the values of the different reward options, one would expect an increase in response latency during choice trials in comparison to sampling trials where the different options are presented separately and successively.

The sequential choice model (SCM) makes a prediction in direct opposition to the DCM. This model explains choice without postulating any explicit deliberation and/or valuation processes. Specifically, when facing different options, animals would consider each option sequentially, not simultaneously, and would decide whether to accept or reject it with no consideration of the other option available. It is hypothesized that such sequential decision process has evolved as an adaptation to the natural 'reward ecology' of most animals. Unlike in laboratory choice settings, successive encounters with different rewards are the rule in natural environments while simultaneous encounters are the exception (*Charnov, 1976*; *Lea, 1982*). Based on this assumption, the SCM proposes that there would be no genuine decision and that preference would be the result of a race-like competition between independent responses. Briefly, during choice, each option would automatically elicit a specific response with a certain latency: the shorter the latency, the more likely the corresponding response will win the race and thus, the more likely the animal will prefer that option. Importantly, due to this race-like selection process, the SCM uniquely predicts that the choice latencies should be shorter, not longer, than the sampling latencies (see *Figure 1—figure supplement 1* for additional information).

Thus, the DCM and SCM models make opposite testable predictions about the differences between choice and sampling latencies (i.e., longer or shorter) (*Table 1*). Interestingly, however, both models make one general common prediction regarding the relationship between sampling latencies and preference, though for different reasons. Both models predict that individuals that respond faster for one option relative to its alternative during sampling trials should also choose the former more frequently than the later during choice trials. In other terms, there should be a positive correlation between individual sampling latencies and individual preferences during choice. According to the SCM, the nature of this relationship should also be predictive since response latencies play a causal role in the establishment of preference in this model (*Shapiro et al., 2008*; *Vasconcelos et al., 2010*). In the DCM, sampling latencies rather represent another measure of the options' relative values.

The goal of the present study is to test these predictions by conducting a systematic retrospective analysis of all cocaine versus saccharin choice experiments that have been conducted in rats in our laboratory over the past 12 years. This allowed us to probe the decision-making mechanisms underlying choice between drug and nondrug rewards as a function of prior training (limited or extended).

## Results

### Lengthening of choice latencies after limited training

We first assessed distributions of response latencies and preference in experiments without prior instrumental training before choice testing (*Figure 1A and B*; *Table 2*, W/O training set). Choice sessions consisted of four sampling trials followed by eight choice trials and separated with 10 min inter-trial intervals. During sampling, each response option was presented twice, alternatively and

**Table 1.** Unique predictions of the different decision-making models.

| Models | Predictions |
| --- | --- |
| Deliberative choice model (DCM) | Sampling latencies < choice latencies |
| Sequential choice model (SCM) | Sampling latencies > choice latencies |

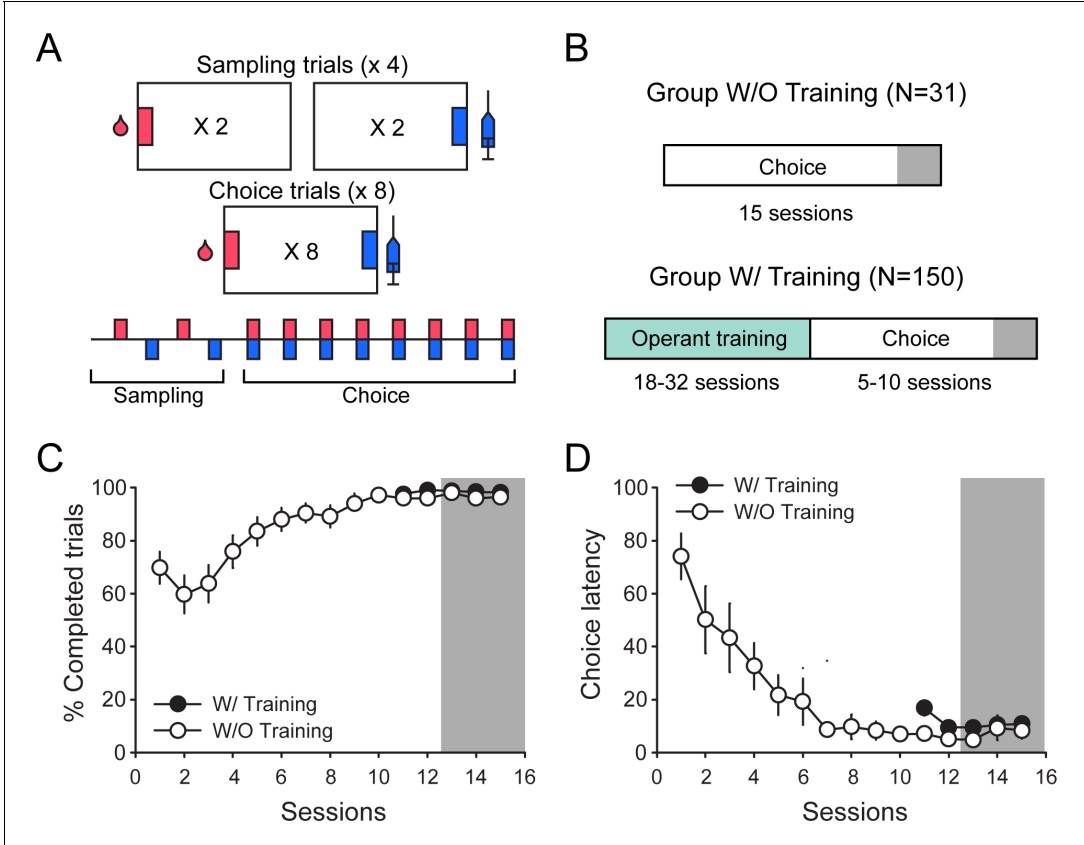

**Figure 1.** Experimental conditions in experiments included in the W/and W/O training sets. (**A**) Diagram of the choice procedure comprising four sampling trials followed by eight choice trials, and separated by 10 min inter-trial intervals. (**B**) Experimental timeline in data sets without (W/O) and with (W/) prior operant training before choice testing. (**C and D**) Mean (± SEM) percentage of completed trials (**C**) and mean (± SEM) choice latency (**D**) across choice sessions in the W/O Training (white circles) and W/Training (black circles) sets. In the W/training set, only the last five choice sessions comprising data for all 150 rats are represented.

The online version of this article includes the following figure supplement(s) for figure 1:

**Figure supplement 1.** Illustration of assumptions and predictions of the sequential choice model (SCM).

sequentially. During choice trials, both response options were presented simultaneously (*Figure 1A*). We assumed that without prior instrumental training, behavior would more likely involve deliberative processes. Indeed, rats' behavior was initially less efficient in the W/O training set with a lower percentage of completed trials (higher number of omission) and longer choice latencies than rats having received prior instrumental training (*Figure 1C and D*; first session, % completed trials: Z = 3.63, p<0.001; choice latency: Z = −4.57, p<0.0001). However, after stabilization of preference with repeated choice testing, the average performance became similar between groups (*Figure 1C and D*; last session; % completed trials: Z = 0.87, p>0.1; choice latency: Z = −0.51, p>0.1). In all analyses, sampling and choice latencies for each response option were analyzed for each individual rat over the last three stable choice sessions (*Figure 1B*; gray area). In total, there were six sampling latencies per option and per rat and 24 choice latencies, with a variable number of responses for cocaine and saccharin, depending on the rat's preference.

Both decision-making models predict that the relative latency of each response option when encountered sequentially (i.e., sampling phase) should predict which one will be selected when encountered simultaneously (i.e., choice phase). That is, the fastest the response for an option during sampling, the more likely this option will be selected during choice. We first tested this prediction by correlating preference with two estimates of the relative response latency for cocaine versus saccharin during sampling trials; the latency ratio (LR) and the proportion of winning latencies (WL). The LR was computed by dividing the mean saccharin sampling latency by the sum of saccharin and

**Table 2.** Summary and conditions of experiments included in the analysis.

| N° Exp | Exp set | N | Prior training | Training session limit | Nb training sessions | Other conditions | Nb choice sessions (FR2) | Selected choice sessions |
|---|---|---|---|---|---|---|---|---|
| 1 | W/O training | 9 | N/A | N/A | 0 | *** | 5 | s13-s15 |
| 2 | W/O training | 11 | N/A | N/A | 0 | *** | 5 | s13-s15 |
| 3 | W/O training | 11 | N/A | N/A | 0 | *** | 5 | s13-s15 |
| | Total | 31 | | | | | | |
| 4 | W/training | 18 | FR1 | 30 rewards/3 hr | 26 | *** | 8 | s32-s34 |
| 5 | W/training | 19 | FR1 | 30 rewards/3 hr | 26 | *** | 8 | s32-s34 |
| 6 | W/training | 11 | FR1 | 30 rewards/3 hr | 18 | 1 month home-cage saccharin access | 10 | s26-s28 |
| 7 | W/training | 22 | FR1/FR2 | 30 rewards/3 hr | 26 | cannula intra-OFC | 10 | s34-s36 |
| 8 | W/training | 21 | FR1 | 30 rewards/3 hr | 23 | cannula intra-OFC | 9 | s30-s32 |
| 9 | W/training | 12 | FR1/FR2 | 30 rewards/3 hr | 21 | *** | 9 | s28-s30 |
| 10 | W/training | 23 | FR1/FR2 | 30 rewards/3 hr | 32 | *** | 5 | s35-s37 |
| 11 | W/training | 9 | FR1 | 30 rewards/3 hr | 19 | *** | 5 | s22-s24 |
| 12 | W/training | 15 | FR1/FR2 | 20 rewards/2 hr | 21 | *** | 5 | s24-s26 |
| | Total | 150 | | | | | | |

cocaine mean latencies. The WL was computed by estimating the probability that each of the six cocaine sampling latencies was shorter (i.e., a win) than each of the six saccharin sampling latencies when compared two by two. LR and WL scores close to one indicate faster cocaine sampling latencies. We found that both LR and WL were positively correlated with the percentage of cocaine choice (*Figure 2A and B*; LR: R = 0.62, p<0.001; WL: R = 0.42, p<0.05). These results suggest that the faster the animals respond for cocaine relative to saccharin during sampling trial, the more they would prefer cocaine.

To determine which decision-making model best predicts choice behavior after limited training, we analyzed the distribution of sampling and choice latencies during saccharin trials in saccharin-preferring (SP) rats. Among the 31 rats included in the analysis, 29 expressed a preference for saccharin (*Figure 2C*). In this subgroup, the distribution of choice latencies was shifted to the right compared to the distribution of sampling latencies (*Figure 2D*). Accordingly, choice latencies were significantly longer than sampling latencies (*Figure 2E*; N = 29, Wilcoxon test: p<0.0001; effect size: −0.99). These results are consistent with the involvement of a deliberative decision-making mechanism.

## Shortening of choice latencies after extended training

We next assessed preference and response latencies during sampling and choice in experiments including prior instrumental training before choice testing (*Figure 1A and B*; *Table 2*, W/training set). Importantly, rats received similar cocaine and saccharin self-administration training on alternate daily sessions before choice testing (*Figure 1B*). We first tested the general common prediction that sampling latencies should correlate with preference and found that, similar to the W/O training set, both LR and WL were positively correlated with the percentage of cocaine choice (*Figure 3A and B*; LR: R = 0.67, p<0.0001; WL: R = 0.64, p<0.0001), indicating that fast responses for cocaine during sampling trials are associated with higher preference for this option during choice.

As expected from prior studies (*Cantin et al., 2010*; *Lenoir et al., 2007*), the majority of rats preferred saccharin with a preference score below 33.3% (SP rats: N = 109, 72.7%; *Figure 3C*). A subset of rats was considered as indifferent with preference scores ranging between 33.3% and 66.6% (IND rats: N = 23, 15.2%) or preferred cocaine with more than 66.6% of cocaine choice (CP rats: N = 19, 12.6%; *Figure 3C*). Interestingly, the SCM not only predicts that preference should be correlated with the relative sampling speed but also proposes that preference could be directly predicted from sampling latencies (*Shapiro et al., 2008*; *Vasconcelos et al., 2010*). To test this hypothesis, we

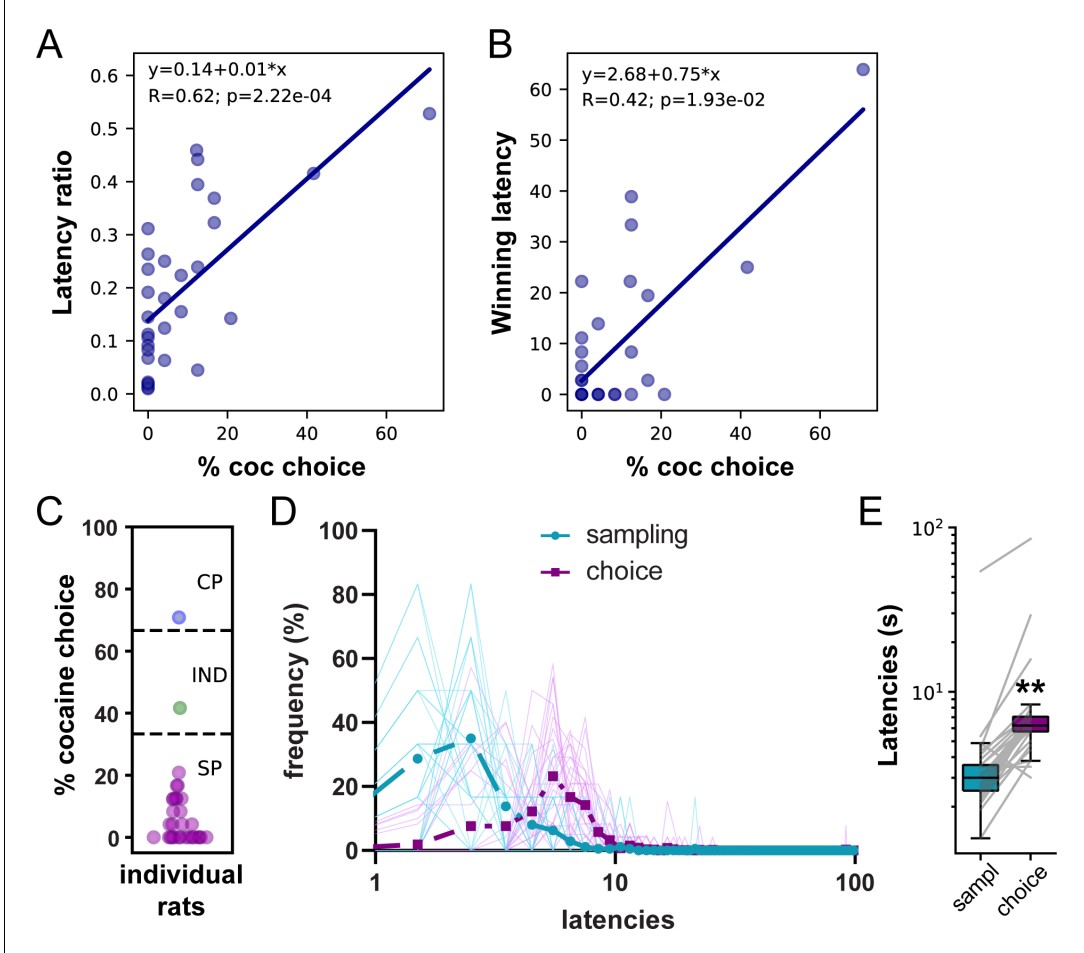

**Figure 2.** Lengthening of saccharin choice latencies compared to saccharin sampling latencies in the W/O training set. (A) Correlation between the latency ratio and the percentage of cocaine choice. (B) Correlation between the winning latency and the percentage of cocaine choice. (C) Distribution of preference scores. Only SP rats (purple circles; N = 29) were considered in the analysis. (D) Distributions of sampling and choice latencies during saccharin trials in SP rats. Transparent solid lines represent distribution of latencies for individual rats. (E) Box plot of saccharin sampling and choice latencies. The box extends from the lower to the upper quartile values with a horizontal line at the median. The whiskers extend from the box at 1.5 times the interquartile range. Gray lines represent mean sampling and choice latencies of individual rats. **p<0.0001.

trained a linear discriminant analysis (LDA) model on the mean cocaine and saccharin sampling latencies on 90% of the data set to classify the preference of individual rats as SP, indifferent (IND), or cocaine-preferring (CP) in the remaining 10%. To avoid any bias resulting from the unbalanced number of SP, IND, and CP rats, we performed the analysis on the same number of rats in each preference group (N = 19), by randomly sampling subjects in the SP and IND groups based on the number of CP rats. This analysis was performed on 50 random selections of SP and IND rats and the performance across all 50 repetitions was averaged to determine the model accuracy. It was possible to predict the preference profile (SP, CP, or IND) from the mean cocaine and saccharin sampling latencies with an accuracy of 53.1 ± 4.4% (*Figure 3D*). A permutation test confirmed that the decoding accuracy significantly departed from chance (i.e. around 33.33%), assessed by shuffling the preference profile (*Figure 3D and E*). This result further validates the first SCM prediction by showing that the preference profile can be predicted above chance, from the options' sampling latencies.

The DCM and SCM make opposite predictions when comparing sampling and choice latencies (*Table 1*). To test these predictions, we compared the individual distributions of sampling and choice latencies for each reward, separately. To avoid any selection bias resulting from saccharin preference in the majority of rats, choice and sampling latencies distributions were compared within-subjects in

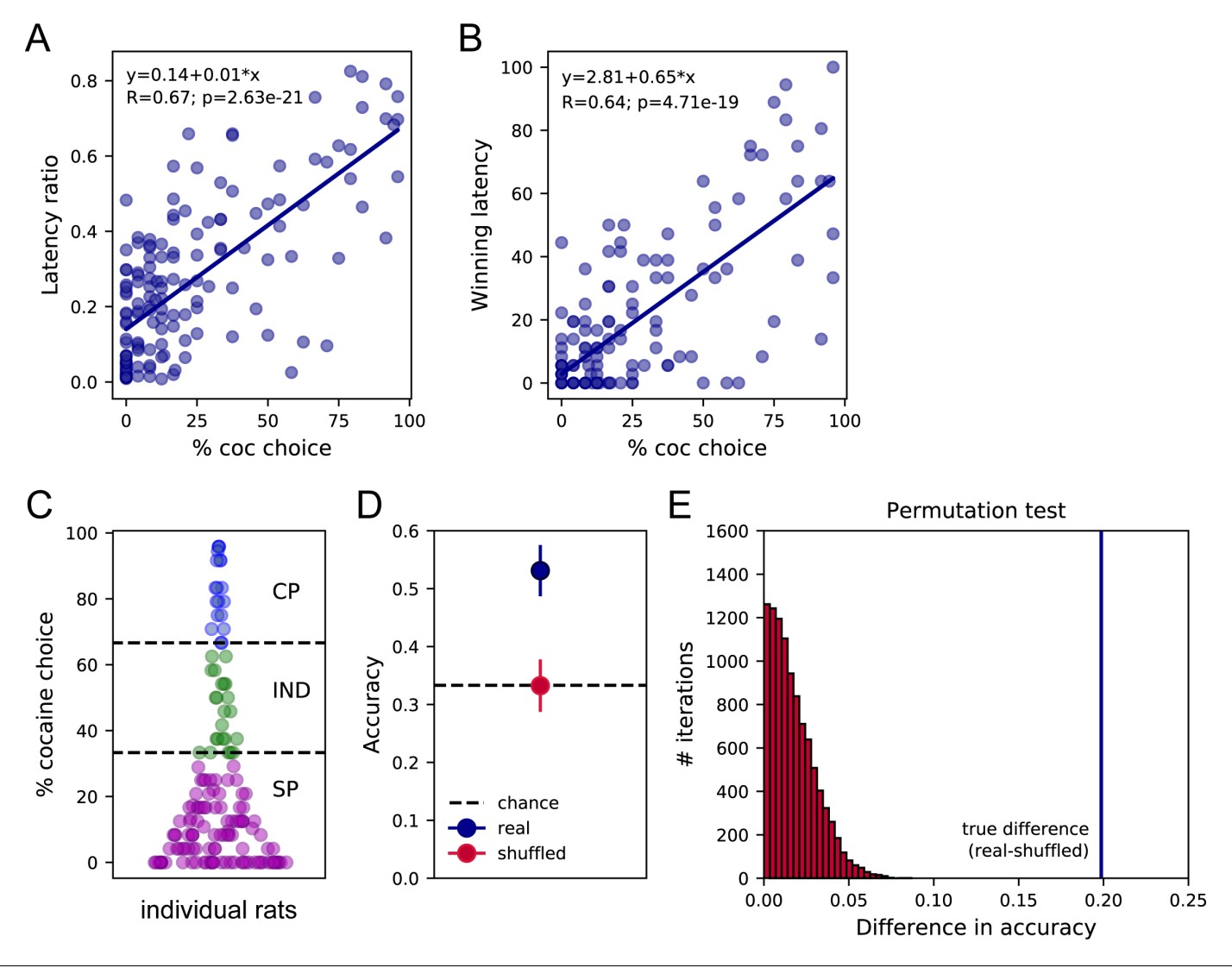

**Figure 3.** Sampling latencies correlate with preference and predict preference profiles. (**A**) Correlation between the latency ratio and the percentage of cocaine choice. (**B**) Correlation between the winning latency and the percentage of cocaine choice. (**C**) Distribution of preference scores and assignment of preference profiles in individual rats. SP: Saccharin-preferring rats, purple circles; IND: Indifferent rats, green circles; CP: Cocaine-preferring rats, blue circles. (**D**) The mean decoding accuracy (± standard deviation) of the preference profile, SP, IND, or CP, of individual rats based on their mean cocaine and saccharin sampling latencies (real – dark blue) is compared with the decoding accuracy expected from chance (shuffled – red; chance level 33.3% – horizontal dashed line). (**E**) Permutation test. The true difference between accuracy scores real-shuffled (vertical blue line) is compared to the distribution of differences in accuracy following permutations with 10,000 iterations. p<0.0001.

different preference groups, separately. Specifically, sampling and choice latencies were only compared for the option that was preferred (saccharin for SP rats, cocaine for CP rats) or chosen (saccharin or cocaine in IND rats) during choice trials.

In SP rats, the distribution of sampling and choice latencies on saccharin trials did not significantly differ, although there was a trend toward longer choice latencies (**Figure 4A**; N = 109; Wilcoxon test: p=0.077; effect size: −0.19). However, in CP rats, we observed a leftward shift in the distribution of latencies during cocaine choice trials compared to cocaine sampling trials (**Figure 4B**). Accordingly, cocaine choice latencies were significantly shorter than cocaine sampling latencies (**Figure 4B**; N = 19, Wilcoxon test: p<0.01; effect size: 0.75). IND rats selected both cocaine and saccharin during choice trials allowing for similar ranges in the number of cocaine and saccharin choice latencies (i.e. between 8 and 16 latencies per reward). When comparing saccharin

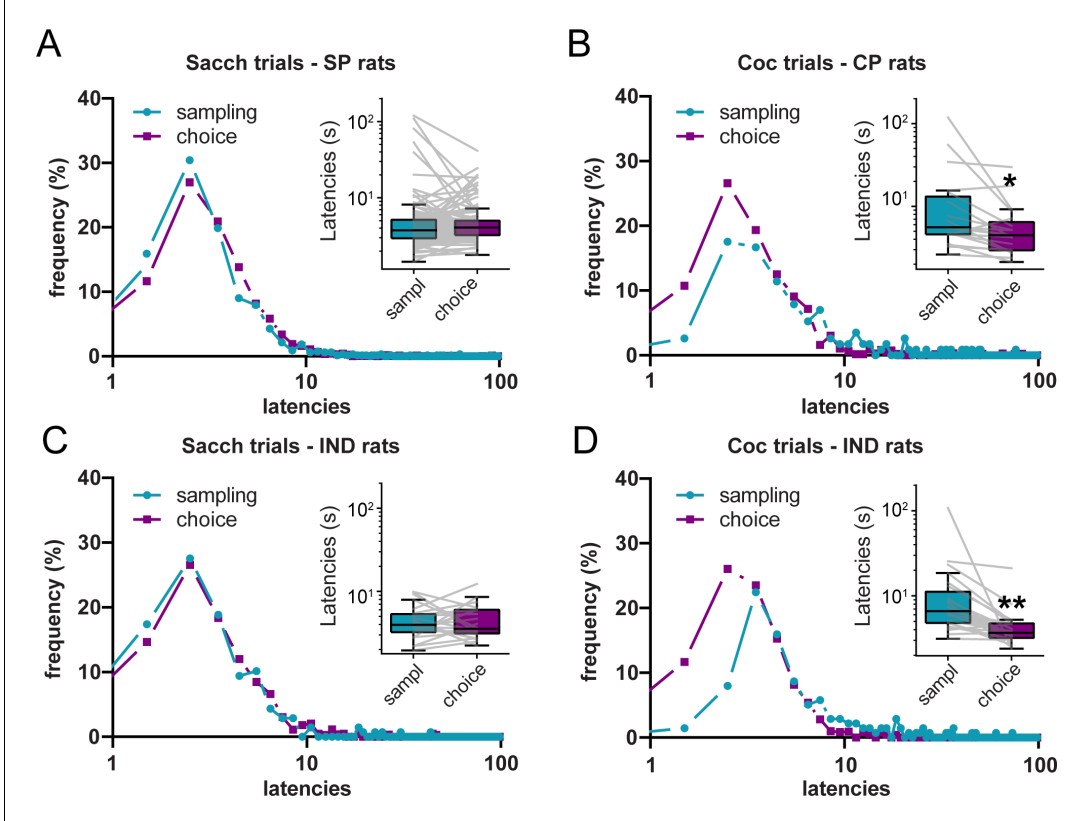

**Figure 4.** Shortening of cocaine choice latencies compared to cocaine sampling latencies. (A–D) Distributions of sampling and choice latencies during saccharin trials in SP rats (A), cocaine trials in CP rats (B), and saccharin (C) or cocaine (D) trials in IND rats. Insets: Box plots of sampling and choice latencies. Boxes extend from the lower to the upper quartile values with a horizontal line at the median. The whiskers extend from the box at 1.5 times the interquartile range. Gray lines represent mean sampling and choice latencies of individual rats. *p<0.01, **p<0.0001.

The online version of this article includes the following figure supplement(s) for figure 4:

**Figure supplement 1.** Similar saccharin sampling and choice latencies when preference for saccharin is habitual.

**Figure supplement 2.** Lengthening of latencies during choice compared to sampling when behavior becomes sensitive to devaluation.

trials, we observed no difference between sampling and choice latencies, similar to saccharin trials in SP rats (*Figure 4C*; N = 23; Wilcoxon test: p>0.9; effect size: −0.0072). However, during cocaine trials, the distribution of choice latencies was shifted to the left compared to the distribution of sampling latencies. Likewise, choice latencies were significantly shorter than sampling latencies (*Figure 4D*; N = 23; Wilcoxon test: p<0.0001; effect size: 0.96). These results partially validate the SCM prediction, since choice latencies were shorter than sampling latencies for the cocaine option but not for the saccharin option. Importantly, although rare, omissions were more frequent during sampling than choice trials, which could have biased the results reported here. However, similar results were found when omission trials were excluded (saccharin trials in SP rats: p=0.06; Cocaine trials in CP rats: p<0.01; saccharin trials in IND rats: p>0.9; cocaine trials in IND rats: p<0.0001).

Although the SCM prediction was only partially validated, the results above suggest that after more extended training, choice behavior does not anymore involve a deliberation over options value, a finding in agreement with prior studies showing that preference for the nondrug reward is under habitual control (*Vandaele et al., 2020*; *Vandaele et al., 2019a*). We next asked whether analysis of choice and sampling latencies in these experiments would confirm the lack of deliberation suggested by rats' preference insensitivity to devaluation of the nondrug reward. In the first experiment, rats were trained to choose between saccharin and cocaine, and saccharin was devaluated by sensory-specific satiety (test 1) or induction of a conditioned taste aversion (test 2). Lever pressing on both levers was not affected by either devaluation methods during the tests conducted in

extinction, indicating that responding for saccharin was habitual. Analysis of saccharin latencies in SP and IND rats during the last three choice sessions immediately preceding devaluation tests indicates no significant difference between mean saccharin sampling and choice latencies (*Figure 4—figure supplement 1*; N = 17, Test 1 Wilcoxon test: p>0.2; effect size: −0.32; Test 2 Wilcoxon test: p>0.2; effect size: −0.29). Although this result does not support the SCM, it confirms that under habitual control, the lack of deliberation can be inferred from comparably short choice and sampling latencies.

In the second experiment, water-restricted rats were offered a choice between water and cocaine. Water was then devalued by satiation. Although water preference was initially insensitive to water devaluation, rats eventually reversed their preference in favor of cocaine following repeated devaluation training (*Vandaele et al., 2019a*). Sampling and choice latencies were compared during baseline privation sessions before and at the end of devaluation training in SP and IND rats. Interestingly, although water sampling and choice latencies did not significantly differ when choice behavior was under habitual control before devaluation training (N = 24; Wilcoxon test: p>0.4; effect size: −0.17), choice latencies were significantly longer at the end of devaluation training (*Figure 4—figure supplement 2*; N = 25; Wilcoxon test: p<0.01; effect size: −0.67), presumably because rats had learned to reengage deliberative decision-making mechanisms to adjust their behavior to repeated changes in water value. This result is in agreement with the DCM prediction (*Table 1*).

## Discussion

The aim of this study was to determine the decision-making processes underlying choice by comparing sampling and choice response latencies, after limited or extended training. An increase in latencies during choice is predicted from the DCM whereas the SCM predicts a shortening of latencies during choice compared to sampling. Here we tested these different predictions in a systematic retrospective analysis of all the choice experiments conducted in the laboratory over the past 12 years. Our analysis shows a lengthening of choice latencies after limited training, suggesting the involvement of a deliberative process. However, when prior training is included before choice testing, we observed similar choice and sampling latencies on saccharin trials and a shortening of choice latencies on cocaine trials. These results are partially consistent with the SCM, but inconsistent with the DCM. Finally, additional analyses in two independent experiments assessing habitual control of choice behavior confirm that the engagement of deliberative or valuation processes can be inferred from the comparison of sampling and choice response latencies.

In agreement with previous experiments (*Monteiro et al., 2020*; *Shapiro et al., 2008*; *Vasconcelos et al., 2010*; *Freidin et al., 2009*; *Vasconcelos et al., 2013*; *Vasconcelos et al., 2015*; *Ojeda et al., 2018*), results from our analysis validated in both experimental sets (with or without prior training) the general common prediction that the relative response latency during sampling trials should correlate with preference during choice trials. Indeed, the parameters of relative latency were both strongly correlated with the percentage of cocaine choices. Furthermore, in agreement with the SCM, which assumes that the relative sampling latency causally determines preference, sampling latencies could predict whether rats preferred cocaine, saccharin, or were indifferent. However, these findings are not sufficient to favor the SCM over the DCM, which also predicts a relationship between sampling latencies and preference. Indeed, according to the DCM, options are compared and selected based on their relative value (*Kacelnik et al., 2011*; *Rangel et al., 2008*; *Rangel and Hare, 2010*) and numerous studies have reported that latencies are inversely related to options' relative value (*Shapiro et al., 2008*; *Bateson and Kacelnik, 1995*; *Killeen and Hall, 2001*; *Lagorio and Hackenberg, 2012*; *Shull et al., 1990*). More generally, it was suggested that response latencies could be used as a sensitive metric for response strength and subjective value (*Shapiro et al., 2008*; *Bateson and Kacelnik, 1995*; *Killeen and Hall, 2001*; *Lagorio and Hackenberg, 2012*; *Shull et al., 1990*). The analysis and comparison of sampling and choice latencies are therefore needed to arbitrate between the DCM and SCM models.

Contrary to all prior studies testing predictions of the SCM (*Kacelnik et al., 2011*; *Monteiro et al., 2020*; *Shapiro et al., 2008*; *Vasconcelos et al., 2010*; *Freidin et al., 2009*; *Vasconcelos et al., 2013*; *Vasconcelos et al., 2015*; *Ojeda et al., 2018*; *Macías et al., 2021*), analysis of experiments without instrumental training prior to choice testing reveals a lengthening of choice latencies compared to sampling latencies. These results are consistent with the DCM which

assumes that processing of information during simultaneous encounters would increase the response latency compared to sequential encounters, due to the time cost of the evaluation process (*Kacelnik et al., 2011*; *Keramati et al., 2011*; *Gold and Shadlen, 2007*; *Rangel et al., 2008*; *Rangel and Hare, 2010*; *Table 1*). The discrepancy with prior findings could result from the relatively long training in instrumental tasks involving discrete trial designs, shown to promote rapid habitual learning (*Vandaele et al., 2017*; *Vandaele et al., 2019b*), or from species differences (*Kacelnik et al., 2011*; *Monteiro et al., 2020*; *Shapiro et al., 2008*; *Vasconcelos et al., 2010*; *Freidin et al., 2009*; *Vasconcelos et al., 2013*; *Vasconcelos et al., 2015*; *Macías et al., 2021*). Indeed, in this analysis, the lengthening of latencies was observed in naïve rats trained for a very limited number of sessions while most studies investigating SCM predictions were conducted in European starlings or pigeons (but see *Ojeda et al., 2018*), after extensive instrumental training.

In the data set with prior instrumental training, the shortening of cocaine response latencies during choice is consistent with the SCM. Indeed, the race-like competition process implies that only the shortest latencies should be expressed during choice trials, the selection of one option excluding the opportunity to select the alternative option (*Figure 1—figure supplement 1*). This cross-censorship between distributions of latencies results in a shortening of choice latencies. This prediction was validated for cocaine but not saccharin trials. The negative result for saccharin trials is however consistent with prior studies testing SCM predictions (*Monteiro et al., 2020*; *Shapiro et al., 2008*; *Vasconcelos et al., 2010*; *Vasconcelos et al., 2013*; *Vasconcelos et al., 2015*; *Ojeda et al., 2018*; *Macías et al., 2021*; *Aw et al., 2012*). Indeed, none of these studies but one (*Ojeda et al., 2018*) has reported a shortening of choice latencies for the preferred option. Demonstrating this decrease in latency is difficult because when animals strongly prefer one option, the distribution of latencies for that option is minimally censored (*Figure 1—figure supplement 1*). Furthermore, the high relative value of this option results in short latencies during sampling, limiting the room for further decrease. Analysis of prior studies conducted in our laboratory indicates that response latencies cannot go below 3 s, likely representing the reaction time.

The floor effect for saccharin response latencies cannot be excluded, but is also likely to occur (*Macías et al., 2021*). In the absence of floor effect, how to explain the similarity between saccharin sampling and choice latencies while there is a shortening of choice latencies on cocaine trial? A less parsimonious hypothesis would be that a shortening of latencies predicted from the SCM is masked by a lengthening of response latencies due to the involvement of deliberative processes. However, this hypothesis is not supported by findings from our devaluation experiments showing that preference for the nondrug reward rapidly becomes habitual (*Vandaele et al., 2020*; *Vandaele et al., 2019a*). In the confirmatory analysis conducted on these experiments, we systematically found similar sampling and choice latencies when responding for the nondrug reward was under habitual control, as indicated by preference insensitivity to devaluation. Although inconsistent with the SCM, this result also invalidates the DCM and confirm that the absence of deliberative or valuation processes can be inferred from the comparison of sampling and choice response latencies. Interestingly, in the second experiment involving choice between water and cocaine, the increased sensitivity to water devaluation was associated with a lengthening of choice latencies compared to sampling latencies. This finding is consistent with the DCM and suggests that across devaluation training, rats are learning to re-engage deliberative processes during choice. Overall, these findings suggest that the DCM and SCM may not be general models of choice, as initially formulated, but could be dynamically engaged to control choice behavior across early and extended training, or following changes in outcomes value or instrumental contingencies.

Although cocaine was the preferred option in CP rats, we were able to observe a shortening of latencies at choice following extended training. Interestingly, this difference was mainly driven by the long sampling latencies since choice latencies were more homogenous between rewards and groups of rats. This finding is in agreement with the SCM, which assumes that while sampling latencies for each option are expressed without cross-censorship from the alternative option, choice latencies are censored by the selection of the alternative option. In CP and IND rats, cocaine sampling latencies are quite long, a finding consistent with prior studies from our lab that remains to be elucidated. However, during choice trials, selection of saccharin censors longer cocaine response latencies, thereby biasing the distribution of cocaine choice latencies in favor of the shortest latencies, more similar to saccharin latencies. Interestingly the gap between cocaine choice and sampling latencies was more pronounced in IND rats compared to CP rats. This result further supports the

SCM, which predicts that trimming of the leftward side of the distribution of latencies would be more pronounced when the overlap between response distributions is stronger (*Kacelnik et al., 2011*; *Shapiro et al., 2008*; *Figure 1—figure supplement 1*). Since rats are indifferent, the drug and nondrug rewards are closer in value and selected with more comparable response latencies during sampling. The stronger cross-censorship between distributions of latencies during choice could have increased the shortening of cocaine choice latencies in this subgroup of rats. However, at indifference, the SCM predicts a shortening of choice latencies for both options (*Macías et al., 2021*). The similarity between saccharin sampling and choice latencies in IND rats is thus unexpected. Given the already low value of latencies during sampling (mean: 4.7 ± 0.49 s; median: 3.97 s), a floor effect cannot be excluded.

Like habit, the automatic selection process of option based on the response latency implied by the SCM is oblivious to the options' value. The SCM assumes that during sequential encounters, each option is processed independently and choice results from a race between the mechanisms generating response latencies for each available option (*Kacelnik et al., 2011*). Thus, although the SCM proposes a mechanism for the selection of responses based on their latency, this model makes no assumption on the valuation process generating distributions of latencies. It is tempting to speculate that the habitual system would process the action value (or 'action policy') while the SCM offers a mechanism by which habit translates in preference in a choice setting between two simultaneous options. This assumption is in agreement with a recent theory suggesting that decision-making does not necessarily require computation of the economic value by the brain (*Hayden and Niv, 2020*). Overall, our analysis suggests that after extended training, responses for the drug and the nondrug rewards are selected via a race-like competition mechanism, without deliberation over options' value, not unlike the habitual preference demonstrated in previous studies (*Vandaele et al., 2020*; *Vandaele et al., 2019a*). Numerous studies show a shift in behavioral control from goal-directed to habitual decision-making processes across extended training (*Holland, 2004*; *Adams, 1982*; *Dickinson and Weiskrantz, 1985*). Thus, our results suggest that rats first engage a goal-directed deliberative strategy with computation and comparison of the options value but, after more extended training, adopt a SCM-like response selection process while their behavior becomes habitual. Interestingly, one would expect similar sampling and choice response latencies for a behavior performed automatically under habitual control and without representation of the options' value. This is in fact what we systematically observed when comparing saccharin latencies in IND and SP rats, after some amount of prior instrumental training. Further research is needed to determine whether comparable sampling and choice latencies could be equated with habit in a choice setting.

A key question raised by our findings is whether the differences between sampling and choice latencies plausibly reflect operation of different decision-making mechanisms (race-like versus deliberative model) or different operations of the same decision-making mechanism. Likewise, it would be interesting to determine whether decision-making mechanisms or operations differ in the groups without and with prior training, when the behavior is presumably goal-directed and habitual, respectively. The interplay between goal-directed and habitual behavior is often described as dichotomous (i.e. based on separate decision-making mechanisms) but there is now a consensus on the need to consider it as a gradient (*Schreiner et al., 2020*), which may involve a single decision-making mechanism with variations in the underlying operations. The diffusion decision model (DDM) or the linear ballistic accumulator (LBA) are both derived from models of human choice reaction time (*Ratcliff et al., 2016*; *Brown and Heathcote, 2008*), and could be applied to our data sets to address this issue. A single decision-making model (LBA or DDM) could explain response latencies in both data sets (with and without prior training) with variations in how the model is operating between sampling and choice and/or between groups, for instance with a change in the starting point of the decision, in the rate of evidence accumulation, or in the threshold at which evidence is translated into choice behavior. Alternatively, behavior in groups with and without prior training could be better explained by different models, indicating that behavior is driven by distinct decision-making mechanisms. Future studies will aim at developing formal modeling of choice behavior to better delineate the decision-making mechanisms or parameters differing between sampling versus choice, limited versus extended training, or as a function of individual preference.

To conclude, this systematic analysis has begun to probe the decision-making processes underlying choice between sweet water and cocaine in rats. It shows that when simultaneously facing these two options, rats first engage a deliberative strategy but after more extended training, rats adopt an

automated SCM-like response selection strategy, in the absence of deliberation. Clearly, more research is needed to dissect the precise decision-making mechanisms underlying choice between drug and nondrug rewards in our discrete-trial choice procedure. However, these results suggest that the comparison of sampling and choice latencies could be used to infer the involvement of deliberative processes during choice.

## Materials and methods

### Subjects

The data analyzed in this study have been obtained from previous experiments conducted in our laboratory over the past 12 years. All experimental subjects were male adult Wistar rats (Charles River, L'Arbresle, France). Rats were housed in groups of two or three and maintained in a temperature-controlled vivarium (23° C) with a 12 hr reverse light–dark cycle. Testing occurred during the dark phase of the cycle and water and food were available ad libitum in all experiments. All experiments were carried out in accordance with institutional and international standards of care and use of laboratory animals [UK Animals (Scientific Procedures) Act, 1986; and associated guidelines; the European Communities Council Directive (2010/63/UE, 22 September 2010) and the French Directives concerning the use of laboratory animals (décret 2013–118, 1 February 2013)]. All experiments have been approved by the Committee of the Veterinary Services Gironde, agreement number B33-063-5.

### Initial operant training

In the first set of experiments, a total of 31 rats were directly tested in the choice schedule without prior operant training (*Table 2*; W/O training set).

In the second set of experiments, a total of 150 rats were first trained several weeks (3-5) under a fixed-ratio (FR) schedule of saccharin and cocaine self-administration on alternate daily sessions, 6 days a week, before choice testing (*Table 2*; W/training set). FR training allowed rats to learn the value of each reward and to associate its delivery with a different response option before choice testing. During FR training, rats had access to a single response option per session with the lever continuously available. On saccharin sessions, lever pressing on the saccharin lever was rewarded by a 20 s access to water sweetened with 0.2% of sodium saccharin delivered in the adjacent drinking cup. During the first 3 s of each 20 s access to sweet water, the drinking cup was filled automatically with sweet water; during the next 17 s, additional volumes of sweet water were obtained on demand by voluntary licking. On cocaine sessions, lever pressing on the alternative lever was rewarded by one intravenous dose of cocaine (0.25 mg delivered over 4 s). For both cocaine and saccharin sessions, reward delivery initiated a concomitant 20 s time-out period signaled by the illumination of the cue-light above the available lever. During the time-out period, responding had no scheduled consequences. Sessions ended after rats had earned a maximum of 20–30 saccharin or cocaine rewards or 2–3 hr had elapsed (*Table 2*).

### Discrete-trials choice protocol

Each daily choice session consisted of 12 trials, spaced by 10 min inter-trials intervals, and divided into two successive phases, sampling and choice. The sampling phase was composed of four sampling trials in which each lever and thus each response option was presented alternatively and sequentially. If rats responded twice within 5 min on the available lever, they were rewarded by the corresponding reward. Reward delivery was signaled by the immediate retraction of the lever and illumination of a cue-light above it. If rats failed to complete the response requirement within 5 min, the lever was retracted until the next trial 10 min later. The choice phase consisted of eight choice trials during which the two response options were presented simultaneously. Specifically, each choice trial began with the simultaneous presentation of both levers S and C and rats could select one of the two levers by responding twice consecutively on it to obtain the corresponding reward. Reward delivery was signaled by the simultaneous retraction of both levers and illumination of the cue-light above the selected lever. If rats failed to respond on either lever within 5 min, both levers were retracted and no cue-light and reward was delivered. Importantly, 10 min inter-trial intervals were set to minimize the direct pharmacological effects of cocaine on subsequent trials and prevent

rats from choosing under the influence of the drug (*Lenoir et al., 2007*; *Vandaele et al., 2016*; *Freese et al., 2018*). A schematic diagram of the forced choice trials procedure can be found in *Ahmed, 2012*; *Lenoir et al., 2013b*.

In the W/O training set, rats were first trained in the discrete-trials choice schedule with a FR1 for 10 sessions before testing with the final FR2 schedule for five sessions.

## Systematic analysis

### Selection of experiments and data analysis

Only choice experiments that were conducted under similar conditions (e.g., initial FR training, conditions of reward delivery, inter-trial intervals, etc.) and that resulted in a stable preference within 5–10 choice sessions (i.e., no increasing or decreasing trend and significant correlation between preference scores over the last three sessions) were included in the present analysis (*Table 2*). The last three sessions of the first experimental phase of choice testing were selected for the analysis to ensure stable preference with low within-subject variability (*Figure 1B*).

Sampling and choice latencies for each response option were analyzed for each individual rat over the last three stable choice sessions. In total, there were six sampling latencies per option and per rat and 24 choice latencies, with a variable number of responses for cocaine and saccharin, depending on the rat's preference. For convenience, performance during choice was expressed in percent of cocaine choices. Response latencies corresponded to the time to complete the FR2 requirement from trial onset (signaled by the lever insertion). When a rat failed to respond within 5 min after trial onset (omission), it was assigned a maximal response latency of 300 s. However, omissions occur rarely.

### Test of the general common prediction

To estimate the relative speed at which the cocaine and saccharin options are selected during sampling trials, we computed for each individual rat the latency ratio (LR) and the proportion of winning latencies (WL). The LR was computed by dividing the mean saccharin sampling latency by the sum of saccharin and cocaine mean latencies. Thus, LR values close to zero indicate faster saccharin sampling latencies while LR values close to one indicate faster cocaine sampling latencies. The WL was computed by estimating the probability that each of the six cocaine sampling latencies was shorter (i.e., a win) than each of the six saccharin sampling latencies when compared two by two. The WL values ranged from 0 (all cocaine latencies are longer than the longest saccharin latency or cocaine never wins the race) to 100% (all cocaine latencies are shorter than the shortest saccharin latency or cocaine always wins the race). Correlation analyses were conducted between these two measures of relative latencies and the percentage of cocaine choices.

To determine whether and to what extent, cocaine and saccharin sampling latencies could predict rat's preference profile, a LDA model (LinearDiscriminantAnalysis from sklearn library in Python) was trained on the mean cocaine and saccharin sampling latencies to classify the preference of individual rats as SP, IND, or CP. Rats were considered as SP, IND, or CP if their preference was below 33.3%, between 33.3% and 66.6% or above 66.6%, respectively. LDA models were trained on 90% of the data set and used to classify rats' preference in the remaining 10% (stratified 10-fold cross-validation with 20 iterations; RepeatedStratifiedKFold from sklearn library in Python). To account for the unbalanced number of SP, IND, and CP rats, we performed the analysis on the same number of rats in each preference group by randomly sampling subjects in the SP and IND groups based on the number of CP rats (N = 19). This analysis was performed on 50 random selections of SP and IND rats and the performance across all 50 repetitions was averaged to determine the model accuracy. The same analysis was conducted with the preference group identities shuffled to determine the model accuracy expected from chance. To assess whether decoding accuracy significantly departed from chance, a permutation test was conducted.

### Comparison of choice and sampling latencies

To test predictions of the different decision-making models, we compared the distributions of sampling and choice latencies for each response option. Note that the number of choice latencies (i.e. max. 24) was larger than the number of sampling latencies (six per option) and that among choice trials, the number of saccharin responses was disproportionally higher than cocaine responses

because of preference for saccharin in the majority of rats. To avoid any selection bias related to differences in preference, choice and sampling responses for saccharin were compared in SP (<33.3% cocaine choice) and IND rats (33.3–66.6% cocaine choice) whereas choice and sampling responses for cocaine were compared in CP (>66.6% cocaine choice) and IND rats. Note that the IND group comprises comparable numbers of cocaine and saccharin choice latencies allowing for analysis of both reward responses in this subgroup of rats.

## Independent confirmatory analysis

To confirm whether the engagement of deliberative process could be inferred from an analysis of response latencies, choice and sampling latencies were compared in two independent experiments testing preference sensitivity to devaluation of the nondrug reward (*Vandaele et al., 2020*; *Vandaele et al., 2019a*). In the first experiment, 20 male Sprague Dawley rats were given a choice between cocaine and saccharin, as described above. We have previously shown that responding for saccharin was not affected by devaluation of saccharin with sensory-specific satiety (test 1) or conditioned taste aversion (test 2) (*Vandaele et al., 2020*). Saccharin sampling and choice latencies across the last three sessions immediately preceding devaluation tests were analyzed. CP rats (>66.6% cocaine choice) were excluded from this analysis.

In the second experiment, 28 male Wistar rats were water-restricted and allowed to choose between water and cocaine, as described above. Water was then devalued with 1 hr water access immediately before the choice session and during every inter-trial intervals of choice sessions (satiation sessions). We have shown that preference was initially insensitive to water devaluation but gradually reversed toward cocaine across repeated cycles of satiation and privation sessions (namely 'devaluation training') (*Vandaele et al., 2019a*). Water sampling and choice latencies were analyzed during the last three privation sessions before devaluation testing and during the last three privation sessions of the devaluation training. CP rats (>66.6% cocaine choice) were excluded from this analysis.

A detailed description of the methods for these two experiments is available in *Vandaele et al., 2020*; *Vandaele et al., 2019a*.

## Statistical analysis

Linear regressions were tested with the Spearman's rank correlation test. Mean sampling and choice latencies were compared within subject using the non-parametric Wilcoxon test. The effect sizes were estimated with the rank-biserial correlation. Following the LDA, a permutation test was conducted to determine whether the decoding accuracy significantly departed from chance. All statistical analyses were conducted on Python.

## Acknowledgements

This work was supported by the French Research Council (CNRS), the Université de Bordeaux, the French National Agency (ANR- 2010-BLAN-1404-01), the Ministère de l'Enseignement Supérieur et de la Recherche (MESR), the Fondation pour la Recherche Médicale (FRM DPA20140629788) and the Peter und Traudl Engelhorn foundation.

## Additional information

### Funding

| Funder | Grant reference number | Author |
| --- | --- | --- |
| Agence Nationale de la Recherche | ANR- 2010-BLAN-1404-01 | Serge H Ahmed |
| Fondation pour la Recherche Médicale | FRM DPA20140629788 | Serge H Ahmed |
| Peter und Traudl Engelhorn Stiftung | | Youna Vandaele |

The funders had no role in study design, data collection and interpretation, or the decision to submit the work for publication.

## Author contributions
Youna Vandaele, Conceptualization, Formal analysis, Visualization, Writing - original draft, Writing - review and editing; Magalie Lenoir, Caroline Vouillac-Mendoza, Karine Guillem, Investigation; Serge H Ahmed, Conceptualization, Resources, Writing - review and editing

## Author ORCIDs
Youna Vandaele (ID) https://orcid.org/0000-0002-8389-8850
Karine Guillem (ID) http://orcid.org/0000-0002-9623-3698

## Ethics
Animal experimentation: All experiments were carried out in accordance with institutional and international standards of care and use of laboratory animals [UK Animals (Scientific Procedures) Act, 1986; and associated guidelines; the European Communities Council Directive (2010/63/UE, 22 September 2010) and the French Directives concerning the use of laboratory animals (décret 2013-118, 1 February 2013)]. All experiments have been approved by the Committee of the Veterinary Services Gironde, agreement number B33-063-5.

## Decision letter and Author response
Decision letter https://doi.org/10.7554/eLife.64993.sa1
Author response https://doi.org/10.7554/eLife.64993.sa2

# Additional files

## Supplementary files
• Transparent reporting form

## Data availability
Data tables and Python codes used to generate figures and conduct statistical analysis have been deposited on Open Science Framework: DOI https://doi.org/10.17605/OSF.IO/BFZJY.

The following dataset was generated:

| Author(s) | Year | Dataset title | Dataset URL | Database and Identifier |
|---|---|---|---|---|
| Vandaele Y, Lenoir M, Vouillac-Mendoza C, Guillem K, Ahmed SH | 2021 | Probing the decision-making mechanisms underlying choice between drug and nondrug rewards in rats | https://osf.io/bfzjy/ | Open Science Framework, 10.17605/OSF.IO/BFZJY |

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
