## [Decision Letter]

**Acceptance summary:**

Decision-making mechanisms underlying choice between drug and nondrug rewards have not yet been delineated. In this manuscript, a retrospective analysis reveals that, in a drug-choice setting, preference in rats appears to be driven initially by deliberative processes, but after extended training shifts to more automatic selection processes.

**Decision letter after peer review:**

Thank you for submitting your article "Probing the decision-making mechanisms underlying choice between drug and nondrug rewards in rats" for consideration by *eLife*. Your article has been reviewed by 3 peer reviewers, and the evaluation has been overseen by Shelly Flagel as the Reviewing Editor and Michael Taffe as the Senior Editor. The following individuals involved in review of your submission have agreed to reveal their identity: Gavan McNally (Reviewer #1); Stephanie Groman (Reviewer #2).

The reviewers have discussed the reviews with one another and the Reviewing Editor has drafted this decision to help you prepare a revised submission.

We would like to draw your attention to changes in our policy on revisions we have made in response to COVID-19 (https://elifesciences.org/articles/57162). Specifically, when editors judge that a submitted work as a whole belongs in *eLife* but that some conclusions require a modest amount of revisions and/or additional new data, as they do with your paper, we are asking that the manuscript be revised to either limit claims to those supported by data in hand, or to explicitly state that the relevant conclusions require additional supporting data.

Summary:

In this manuscript the authors perform a retrospective analysis in attempt to delineate the role of goal-directed versus habitual mechanisms underlying choice between drug and non-drug rewards. Specifically, the authors utilized data generated in their laboratory to assess cocaine-versus-saccharin choice following limited and extended training paradigms. A sequential choice model was used to assess the prediction that increased latencies during choice reflects goal-directed control; whereas no change in latencies reflects habitual control. Based on this model, the authors report that rats engage in goal-directed control after limited training, and adopt more habitual responding after extended training. The authors conclude that the sequential choice model is specific to habitual choice.

We know already from decades of work that choice response times are almost always log-normally distributed (humans, non-human primates, rodents). The question here is whether differences in the mean and dispersion of these distributions can be used to derive insights into nature of the decision-making mechanism – a deliberative comparison versus a race model – and how this may differ for rats that prefer cocaine over saccharin and how this might be altered by more extended training. These questions are framed in terms of the differences between goal-directed and habitual behavior, which may be less compelling than considering the significance of the data in their own right.

While the Reviewers appreciate the approach and conceptual framework described in this manuscript, they are all in agreement that additional data and analyses are needed to better support the claims surrounding goal-directed versus habitual control of reward-seeking behavior. For example, an independent evaluation of whether the target behavior is in fact goal-directed or habitual seems necessary to support such claims. Additional concerns raised by the Reviewers and suggestions for improvement our included below.

Essential revisions:

1. Much, if not all, of the absolute differences between latencies in sample and choice phases appear to be carried by the sample rather than the choice phase. Choice latencies for cocaine preferring rats, saccharin preferring rats, and the indifferent rats are all very similar. In contrast, the sampling latencies for cocaine preferring rats and the indifferent rats are longer. Why is this the case? The authors seem more concerned with the choice side of the experiment being different, not the sample phase. Is this predicted by the models being tested? Why would an SCM-like model would predict the difference being in the sample phase. Either way, the authors should be clearer about where the difference is expected to lie and why the sample phase is so obviously different in some conditions but the choice phase so similar.

2. The major issue with this manuscript is whether the differences between response latencies in the sample versus choice phases plausibly reflect operation of *different* decision making mechanisms (race model versus deliberative processing) or different operation of the *same* decision-making mechanism. The answer is difficult to derive from the data and modelling provided. The authors frame the differences in response time as being uniquely predicted or explained by different forms of choice. The models that the authors are using are closely linked to, and intellectually derived from, models of human choice reaction time. The most successful of these models are the diffusion model (DDM) (Ratcliff, R., Smith, P.L., Brown, S.D., and McKoon, G. (2016). Diffusion Decision Model: Current Issues and History. Trends in Cognitive Sciences 20, 260-281) and the linear ballistic accumulator (LBA) (Brown, S.D., and Heathcote, A. (2008). The simplest complete model of choice response time: linear ballistic accumulation. Cognitive Psychology 57, 153-178.2008).

Even though the DDM and LBA adopt different architectures to each other (but the same architectures as those in Supp Figure 1A), they are intended to explain the same data. Of relevance, the same model (a DDM or an LBA) can explain differences in both the response distribution and the mean response time via changes in the starting point of evidence accumulation, rate of evidence accumulation, and/or the boundary or threshold at which evidence is translated into choice behavior. So, for either a difference accumulator model (DDM) or a race model (LBA), the difference between sampling and choice performance could reflect changes in how the model is operating between these two phases, including a change in the starting point of the decision [bias], a change in rate of accumulation [evidence], a change in threshold [caution] or collapsing boundary scenario, rather than reflecting operation of a completely different decision-making mechanism.

Another option could be to acknowledge this possibility and discuss it. e.g., does it really matter if it is a qualitatively different decision-making process or different operation of the same decision-making mechanism? The action-habit distinction doesn't live or die by reaction/response time data, this distinction is almost certainly far less absolute than often portrayed in the addiction literature, and it is generally intended as an account of what is learned rather than an account of how that learning is translated into behaviour (even if an S-R mechanism provides an account of both). Response time data likely reflect something different- about how what has been learned is translated into behaviour. The third, marginally more difficult but more interesting option, would be to explore these issues formally and to move beyond simple descriptive or LDA analyses of response time distributions. The LBA has a full analytical solution and there are reasonable approximations for the DDM. Formal modelling of choice response times (e.g., Bayesian parameter estimation for a race model or DDM) could indicate whether a single decision-making mechanism (LBA or DDM or something else) can explain response times under both sample and choice conditions or not. This is a standard approach in cognitive modelling. This would be compelling if it showed the dissociation the authors argue – i.e. one model cannot be fit to both sample and choice datasets for all animals. However, if one model can expain both, then formal modelling would show which decision making parameters change between the sample and choice conditions for cocaine v sacc v ind animals to putatively cause the differences in response times observed. Either way, more formal modelling would provide a platform towards identification of those specific features of the decision-making mechanisms that are being affected.

3. The use of response times between sampling and choice behavior as a proxy for assessing habitual vs. goal-directed behaviors is quite clever and the predictions are straightforward. However, the authors do not provide any data to validate this metric. For example, are there data in these animals using more traditional tests of habit (e.g., devaluation) that could be used to support their argument that behavior is habitual when sampling latencies are equivalent to choice latencies? Even using devaluation procedures there is variation across subjects, so perhaps the authors could correlate the difference score with responses during devaluation? These additional data would better support the utility of this task and analytic approach.

4. Most animals prefer saccharin and perform more saccharin-seeking responses; thus, the outcomes are not matched. Would cocaine RT become faster with more equivalent practice? This model should be applied perhaps in the first instance to responding for two natural rewards, where devaluation could confirm the nature of control and directly test the ideas from the model.

5. The stated aim of the study was to determine the decision-making processes underlying choice, but the data presented do not directly address this aim. It may predict choice, but not how that choice is arrived at? To address this, the model should be applied perhaps in the first instance to responding for two natural rewards, where devaluation could confirm the nature of control and directly test the ideas from the model.

6. The authors state that they can't rule out a floor effect, but this is an important possibility that if true would completely change the interpretation of the findings. More discussion of the implications is warranted.

[Editors' note: further revisions were suggested prior to acceptance, as described below.]

Thank you for submitting your article "Probing the decision-making mechanisms underlying choice between drug and nondrug rewards in rats" for consideration by *eLife*. Your article has been reviewed by 2 peer reviewers, and the evaluation has been overseen by Shelly Flagel as the Reviewing Editor and Michael Taffe as the Senior Editor. The following individual involved in review of your submission has agreed to reveal their identity: Stephanie Mary Groman (Reviewer #2).

The reviewers have discussed their reviews with one another, and the Reviewing Editor has drafted this to help you prepare a revised submission. Although we generally try to avoid multiple rounds when the authors are unwilling to adjust their claims that reviewers evaluated as insufficiently supported, we believe another opportunity to revise in light of these significant concerns is justified here.

Essential revisions:

1. While the authors were generally responsive to prior concerns, additional revisions are needed to temper the claims surrounding goal-directed versus habitual control of reward-seeking behavior. As noted by all of the reviewers previously, the reported procedures do not allow the unequivocal identification of the psychological bases of action vs. habit. Thus, in lieu of additional data or analyses, the authors need to re-frame the manuscript and moderate their conclusions. The reviewers believe that the importance of these data is independent of the action vs. habit distinction, and feel that the impact of the manuscript will be greater if alternative interpretations are further considered.

Reviewer #1:

I remain of the view that this is a fascinating and potentially important manuscript. However, I am remain as unsure as to what it really means. There are differences in RTs, but as noted by all reviewers previously, the procedures simply do not allow unequivocal identification of the psychological bases for these differences (action, habit; other). This is somewhat unfortunate as the manuscript is framed in terms of these differences. As indicated previously, I think the importance of these data is independent of the kinds of action/habit distinctions the authors are trying to draw. Likewise, the modelling is novel but does not really solve the problem of whether different decision mechanisms for choice are involved or whether the same mechanism is operating in different ways. For these reasons, I find myself in much the same place after reading the revision as I did the original manuscript: intrigued, seeking to be supportive, but quite unsure as to what it means.

Reviewer #2:

The authors have addressed my previous comments and I have no additional concerns. This is an excellent manuscript with fantastic results – well done.

---

## [Author Response]

Essential revisions:1. Much, if not all, of the absolute differences between latencies in sample and choice phases appear to be carried by the sample rather than the choice phase. Choice latencies for cocaine preferring rats, saccharin preferring rats, and the indifferent rats are all very similar. In contrast, the sampling latencies for cocaine preferring rats and the indifferent rats are longer. Why is this the case? The authors seem more concerned with the choice side of the experiment being different, not the sample phase. Is this predicted by the models being tested? Why would an SCM-like model would predict the difference being in the sample phase. Either way, the authors should be clearer about where the difference is expected to lie and why the sample phase is so obviously different in some conditions but the choice phase so similar.

This is an interesting issue that we now better explain in the Discussion (p10, lines 24-32). Briefly, the fact that latency differences mainly appear during the sampling phase and not during the choice phase is a direct consequence of the response latency-based censorship among options hypothesized by the SCM. Indeed, while sampling latencies for each option are expressed without censorship by the alternative option, choice latencies are systematically censored by selection of the alternative option, thereby biasing their distribution in favor of the shortest choice latencies. This response latency-based censorship also inevitably results in a homogenization of choice latencies across preference groups.

Now, the differences in sampling latencies between the different groups of rats (i.e., saccharin-preferring rats, cocaine-preferring rats, and indifferent rats) were in fact due to the type of options sampled that were considered for analysis, as now better explained in the text (p7, lines 5-7). Regardless of the groups, sampling latencies for cocaine were longer than sampling latencies for saccharin. In Figure 3 (now Figure 4), this difference between options appears as a difference between groups because for the purpose of our analysis, we only compare the sampling and choice latencies of the option that is preferred (saccharin for saccharin-preferring rats, cocaine for cocaine-preferring rats) or chosen (saccharin or cocaine in indifferent rats) during choice trials. The reason why sampling latencies for cocaine are longer than sampling latencies for saccharin has not been elucidated yet. To paraphrase Newton, we “feign no hypotheses” here.

2. The major issue with this manuscript is whether the differences between response latencies in the sample versus choice phases plausibly reflect operation of different decision making mechanisms (race model versus deliberative processing) or different operation of the same decision-making mechanism. The answer is difficult to derive from the data and modelling provided. The authors frame the differences in response time as being uniquely predicted or explained by different forms of choice. The models that the authors are using are closely linked to, and intellectually derived from, models of human choice reaction time. The most successful of these models are the diffusion model (DDM) (Ratcliff, R., Smith, P.L., Brown, S.D., and McKoon, G. (2016). Diffusion Decision Model: Current Issues and History. Trends in Cognitive Sciences 20, 260-281) and the linear ballistic accumulator (LBA) (Brown, S.D., and Heathcote, A. (2008). The simplest complete model of choice response time: linear ballistic accumulation. Cognitive Psychology 57, 153-178.2008).Even though the DDM and LBA adopt different architectures to each other (but the same architectures as those in Supp Figure 1A), they are intended to explain the same data. Of relevance, the same model (a DDM or an LBA) can explain differences in both the response distribution and the mean response time via changes in the starting point of evidence accumulation, rate of evidence accumulation, and/or the boundary or threshold at which evidence is translated into choice behavior. So, for either a difference accumulator model (DDM) or a race model (LBA), the difference between sampling and choice performance could reflect changes in how the model is operating between these two phases, including a change in the starting point of the decision [bias], a change in rate of accumulation [evidence], a change in threshold [caution] or collapsing boundary scenario, rather than reflecting operation of a completely different decision-making mechanism.Another option could be to acknowledge this possibility and discuss it. e.g., does it really matter if it is a qualitatively different decision-making process or different operation of the same decision-making mechanism? The action-habit distinction doesn't live or die by reaction/response time data, this distinction is almost certainly far less absolute than often portrayed in the addiction literature, and it is generally intended as an account of what is learned rather than an account of how that learning is translated into behaviour (even if an S-R mechanism provides an account of both). Response time data likely reflect something different- about how what has been learned is translated into behaviour.

It would indeed be very interesting to determine whether the differences in response latencies during sampling versus choice trials reflect operation of different decision-making mechanisms or different operations of the same mechanism. We agree with the reviewer that our analysis falls short of addressing this question, but we feel that it is beyond the scope of our study. In fact, the sequential choice model is oblivious to the decision-making mechanisms underlying the selection of a particular option. However, as suggested by the reviewer, we now raise this question in the Discussion and discuss in some details how the diffusion and linear ballistic accumulator models could help to tackle it (p11 lines 12-30). We also acknowledge the limitations of the action-habit dichotomy and, thus, discuss the possibility of a gradient between action and habit that may involve a single decision-making mechanism with a change in underlying operations (p11, lines 12-30).

The third, marginally more difficult but more interesting option, would be to explore these issues formally and to move beyond simple descriptive or LDA analyses of response time distributions. The LBA has a full analytical solution and there are reasonable approximations for the DDM. Formal modelling of choice response times (e.g., Bayesian parameter estimation for a race model or DDM) could indicate whether a single decision-making mechanism (LBA or DDM or something else) can explain response times under both sample and choice conditions or not. This is a standard approach in cognitive modelling. This would be compelling if it showed the dissociation the authors argue – i.e. one model cannot be fit to both sample and choice datasets for all animals. However, if one model can expain both, then formal modelling would show which decision making parameters change between the sample and choice conditions for cocaine v sacc v ind animals to putatively cause the differences in response times observed. Either way, more formal modelling would provide a platform towards identification of those specific features of the decision-making mechanisms that are being affected.

This is an interesting perspective for future research that requires collaboration with expert modelers. Formal modelling on our dataset could provide compelling supplemental information on the decision-making mechanisms or parameters differing between sampling versus choice, limited versus extended training or as a function of individual preference. We now describe this perspective of our study in the Discussion (p11, lines 22-30).

3. The use of response times between sampling and choice behavior as a proxy for assessing habitual vs. goal-directed behaviors is quite clever and the predictions are straightforward. However, the authors do not provide any data to validate this metric. For example, are there data in these animals using more traditional tests of habit (e.g., devaluation) that could be used to support their argument that behavior is habitual when sampling latencies are equivalent to choice latencies? Even using devaluation procedures there is variation across subjects, so perhaps the authors could correlate the difference score with responses during devaluation? These additional data would better support the utility of this task and analytic approach.

As suggested by the reviewer, we analyzed choice and sampling latencies in two independent experiments in which the nondrug reward was devalued (Vandaele et al., 2019 and 2020). When preference was insensitive to devaluation of the nondrug reward, choice and sampling latencies were similar, whereas a lengthening of choice latencies compared to sampling latencies was observed when preference became sensitive to devaluation after repeated devaluation training. These results confirm our model and have now been included in the manuscript with an additional main figure (Figure 5) and a supplemental figure (Supplemental Figure 2).

Analysis of correlation between sensitivity to devaluation and the difference score between sampling and choice latencies did not provide consistent results in the different stages and experiments. This negative result could be explained by the low inter-individual variability in the sensitivity to devaluation when choice behavior is habitual.

4. Most animals prefer saccharin and perform more saccharin-seeking responses; thus, the outcomes are not matched. Would cocaine RT become faster with more equivalent practice? This model should be applied perhaps in the first instance to responding for two natural rewards, where devaluation could confirm the nature of control and directly test the ideas from the model.

Although most animals prefer saccharin over cocaine, they received similar initial operant training with these two options before choice testing (p6, lines 14-15). Thus, it is unlikely that long response latencies for cocaine results from less operant practice with this reward. In fact, response latencies for cocaine remain longer than response latencies for saccharin even in rats that receive largely more operant practice with cocaine than with saccharin before choice testing (see Lenoir et al. 2007; see also, our response to comment 1 above). We agree, however, with the reviewer that our conclusions would benefit from an independent confirmation of our model in a choice situation between two nondrug rewards. This perspective for future research is now acknowledged in the Discussion (p10, lines 18-22).

5. The stated aim of the study was to determine the decision-making processes underlying choice, but the data presented do not directly address this aim. It may predict choice, but not how that choice is arrived at? To address this, the model should be applied perhaps in the first instance to responding for two natural rewards, where devaluation could confirm the nature of control and directly test the ideas from the model.

Our study does show that the goal-directed versus habitual nature of preference can be inferred from the comparison of choice and sampling latencies. Thus, the stated aim of our study, consisting in determining the decision-making mechanisms (habit, goal-directed, SCM) underlying choice between a drug and a nondrug reward is addressed. We now report additional analysis of two independent experiments assessing preference sensitivity to devaluation that further supports our main conclusions (see response to comment 3). We also acknowledge the value of additional research in a choice situation involving two nondrug rewards (see our response to comment 4). However, our main goal here was to investigate decision-making mechanisms underlying choice between drug and nondrug rewards. Indeed, it remains difficult to determine whether preference for the drug is also under habitual control, mainly because there is no effective method of drug reward devaluation in animals, particularly for drugs administered intravenously. Thus, our study provides an alternative approach to drug devaluation to infer the goal-directed versus habitual nature of choice.

6. The authors state that they can't rule out a floor effect, but this is an important possibility that if true would completely change the interpretation of the findings. More discussion of the implications is warranted.

As suggested by the reviewer, we are now more thoroughly discussing this issue in the Discussion and its potential implications for our conclusions (p10 lines 1-7). Overall, we conclude that a floor effect is real, but this does not compromise the interpretation of our findings.

[Editors' note: further revisions were suggested prior to acceptance, as described below.]

Essential revisions:1. While the authors were generally responsive to prior concerns, additional revisions are needed to temper the claims surrounding goal-directed versus habitual control of reward-seeking behavior. As noted by all of the reviewers previously, the reported procedures do not allow the unequivocal identification of the psychological bases of action vs. habit. Thus, in lieu of additional data or analyses, the authors need to re-frame the manuscript and moderate their conclusions. The reviewers believe that the importance of these data is independent of the action vs. habit distinction, and feel that the impact of the manuscript will be greater if alternative interpretations are further considered.

We re-framed the manuscript under a broader perspective and tempered our conclusions about the distinction between goal-directed and habitual decision-making mechanisms based on the analysis of response latencies.

Reviewer #1:I remain of the view that this is a fascinating and potentially important manuscript. However, I am remain as unsure as to what it really means. There are differences in RTs, but as noted by all reviewers previously, the procedures simply do not allow unequivocal identification of the psychological bases for these differences (action, habit; other). This is somewhat unfortunate as the manuscript is framed in terms of these differences. As indicated previously, I think the importance of these data is independent of the kinds of action/habit distinctions the authors are trying to draw. Likewise, the modelling is novel but does not really solve the problem of whether different decision mechanisms for choice are involved or whether the same mechanism is operating in different ways. For these reasons, I find myself in much the same place after reading the revision as I did the original manuscript: intrigued, seeking to be supportive, but quite unsure as to what it means.

We understand the reviewer’s concern and appreciate his/her willingness to be supportive. In response to this last comment, we tried to re-frame the manuscript to test the validity of the deliberative choice model (DCM) and the sequential choice model (SCM) as a function of prior training. In this new revision, we are not pretending to tease apart habit from action on the basis of response latencies. Instead, we tempered our conclusions and highlighted in the Discussion the similitudes between habit and the SCM, on one hand, and between goal-directed behavior and the DCM, on the other hand (p10 lines 17-29). Overall, we now suggest that the comparison of sampling and choice latencies could be used to infer the involvement of deliberative processes during drug versus nondrug choice.